# Enhancing Reliability across Short and Long-Form Question Answering via Reinforcement Learning

## Abstract

While reinforcement learning has unlocked unprecedented complex reasoning in large language models, it has also amplified their propensity for hallucination, creating a critical trade-off between capability and reliability. This work confronts this challenge by introducing a targeted RL framework designed to mitigate both intrinsic and extrinsic hallucinations across short and long-form question answering. We address extrinsic hallucinations (flawed internal knowledge) by creating a novel training set from open-ended conversions of TriviaQA. Concurrently, we tackle intrinsic hallucinations (unfaithfulness to context) by leveraging long-form texts from FineWeb in a fact-grounding reward scheme. To further bolster reliability, our framework explicitly rewards the model for refusing to answer unanswerable questions, thereby cultivating crucial cautiousness. Extensive experiments demonstrate that our methodology yields significant performance gains across a diverse suite of benchmarks, substantially reducing both hallucination types. Ultimately, this research contributes a practical framework for resolving the critical tension between advanced reasoning and factual trustworthiness, paving the way for more capable and reliable large language models.

## 1 Introduction

Recent advancements in reinforcement learning (RL) have empowered large language models (LLMs) to exhibit longer chain-of-thought (CoT) capabilities, significantly enhancing their complex reasoning abilities (Guo et al., 2025; Jaech et al., 2024). However, this progress comes at a cost, as these models show higher hallucination rates than their base counterparts (OpenAI, 2025; Yao et al., 2025). This heightened hallucination rate in RL-driven models may stem from an avalanche effect where long CoT chains lead to irreversible error accumulation. Additionally, existing research has prioritized reasoning enhancement over hallucination mitigation.

Hallucinations are typically categorized as intrinsic or extrinsic (Ji et al., 2023). Extrinsic hallucinations are often defined as errors in a model's internal knowledge and are frequently confused with "factuality" due to varying definitions across different studies (Bang et al., 2025; Yao et al., 2025). In this work, we define extrinsic hallucinations broadly to include both the generation of entirely fabricated knowledge and relational fallacies (e.g., temporal inaccuracies). In contrast, intrinsic hallucinations occur when a model fails to use knowledge explicitly provided by the user, such as not following instructions or ignoring given reference material. While some studies have used RL to address hallucinations (Yang et al., 2025b; Song et al., 2025; Ren et al., 2025), they often rely on highly restrictive methods, such as generating short-form outputs or simply having the model refuse to answer. These approaches are limited to mitigating internal knowledge or factuality issues, leaving a notable research gap in addressing extrinsic hallucinations—a growing concern in the wake of RL-driven LLM advancements.

In this work, we categorize question answering (QA) tasks into two main types: short-form QA and long-form QA. For short-QA, which includes tasks focused on factuality and unanswerable questions, we designed a novel RL reward method. Following the approach of prior work (Yang et al., 2025b; Song et al., 2025; Ren et al., 2025), our method successfully improves model performance while simultaneously enhancing reliability.

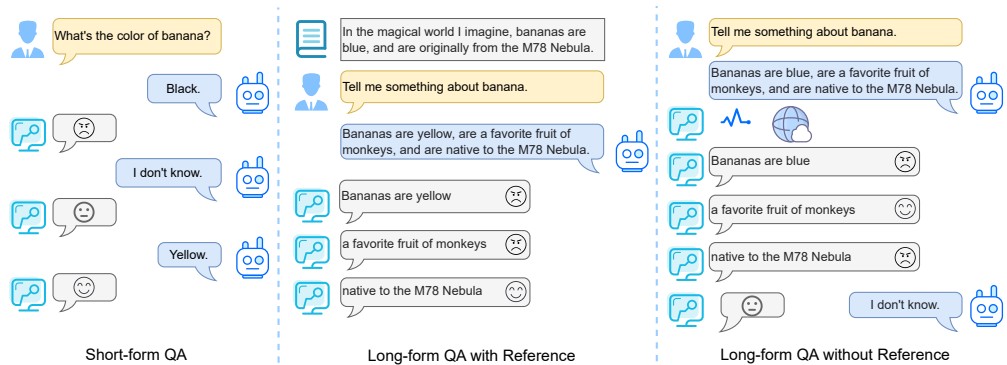

Figure 1: In this work, we classify Hallucinations into three types. In Short-form QA, answers are directly verified. In Long-form QA with reference, claims are checked against the provided text to assess for intrinsic hallucinations. In Long-form QA without reference, claims are checked against search results to assess for extrinsic hallucinations.

For long-form QA, we constructed a training set spanning two distinct scenarios: with and without reference content. For the scenario with reference content, we adopted the evaluation method of FACTS Grounding (Jacovi et al., 2025), retrieving 2,000 high-quality data samples from Fineweb (Penedo et al., 2024). We then used LLMs to generate targeted questions for each sample. For the scenario without reference content, we retrieved content from TriviaQA (Joshi et al., 2017) search results and converted the questions into an open-ended format. Experiments demonstrate that our method can significantly enhance the model's reliability on the evaluation dataset.

We also analyzed several factors that might affect performance. Regarding CoT, we experimented with three supervision approaches: full supervision, summarizing CoT after reasoning, and no supervision at all. We found that supervising CoT did not lead to significant improvements in long-form performance. We also observed a tendency for the model to reduce its output length when answering long-form questions. While we designed three methods to address this issue, we found that each involved certain trade-offs.

Our primary contributions in this work can be summarized as follows:

**A Novel RL Framework for Broad-Spectrum Hallucination Mitigation:** We propose a new reinforcement learning framework designed to mitigate both intrinsic and extrinsic hallucinations across short-form and long-form QA. Our approach addresses a critical research gap left by prior methods that were often restricted to short-form factuality or simple refusal behaviors.

**New Training Datasets for Long-Form QA:** We construct and introduce a diverse training dataset for long-form QA, featuring two distinct scenarios to target different types of hallucinations. The first scenario uses reference-grounded content derived from Fineweb to improve faithfulness (mitigating intrinsic hallucinations), while the second uses open-ended questions adapted from TriviaQA to target the model's internal knowledge (mitigating extrinsic hallucinations).

**Comprehensive Empirical Analysis and Practical Insights:** We conduct a detailed analysis of several factors in the RL training process, providing practical insights for the field. Our findings demonstrate that:

- Direct supervision of CoT provides negligible performance gains for long-form tasks relative to its high computational cost.
- A fundamental trade-off exists between encouraging detailed responses and maintaining factual accuracy, for which we design and evaluate several distinct countermeasures.

## 2 RELATED WORKS

**Hallucination Benchmarks** Hallucinations in large language models are broadly categorized as *intrinsic* (unfaithfulness to a provided source) and *extrinsic* (factual errors from flawed parametric

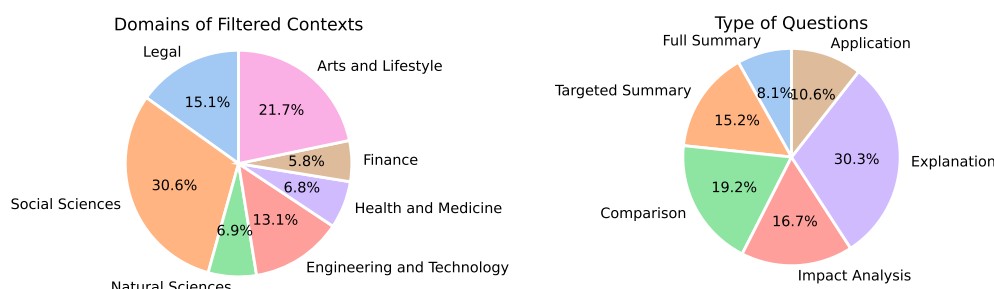

Figure 2: Composition of the training data constructed from the FineWeb dataset. The left chart illustrates the distribution of subject domains for the filtered source contexts, while the right chart shows the distribution of the types of questions generated based on those contexts.

knowledge) (Ji et al., 2023; Huang et al., 2025). Accordingly, evaluation has evolved from early benchmarks assessing short-form factuality (Joshi et al., 2017; Lin et al., 2021; Wei et al., 2024a) to modern, LLM-aided assessments of long-form faithfulness Jacovi et al. (2025) and factuality (Wei et al., 2024b; Min et al., 2023).

**Post-training For Hallucination**   Post-training methods to mitigate hallucinations primarily involve either alignment techniques like Supervised Fine-Tuning and Direct Preference Optimization (DPO) using curated datasets (Rafailov et al., 2023; Lin et al., 2024), or grounding outputs in external knowledge via Retrieval-Augmented Generation (RAG) (Yu et al., 2023; Ram et al., 2023).

**Online Reinforcement Learning**   Online Reinforcement Learning (RL) has become a prominent alignment strategy, pioneered by OpenAI's O1 model using Proximal Policy Optimization (PPO) (Schulman et al., 2017; Jaech et al., 2024) and now widely adopted with similar methods like GRPO (Shao et al., 2024) in models such as DeepSeek-R1 (Guo et al., 2025) and other leading LLMs (Yang et al., 2025a; Team et al., 2025). However, a notable side effect of RL-based training is the facilitation of extended chain-of-thought reasoning, which, while beneficial for complex tasks, correlates with an increased prominence of hallucinations (Yao et al., 2025).

## 3 REINFORCEMENT LEARNING FOR HALLUCINATION MITIGATION

### 3.1 TRAINING DATA SYNTHESIS

To comprehensively enhance model reliability, our training corpus incorporates three distinct data formats: short-form QA, long-form QA with references, and long-form QA without references.

**Short-Form QA**   Our short-form QA data is an aggregation of several sources: the open-source TriviaQA training set (Joshi et al., 2017); a synthetic training set of unanswerable mathematical questions (Song et al., 2025); and a set of answerable mathematical problems from DeepScaler (Luo et al., 2025). To improve the model's ability to recognize unsolvable queries, 25

**Long-Form QA with References**   We constructed a dataset for long-form QA with a reference via a structured generation process. First, we selected high-quality texts from the FineWeb dataset with lengths ranging from 32K to 80K characters (about 8K to 20K tokens). Subsequently, we ask for LLM to generate one question for each text, designed to span six distinct categories: impact analysis, specific content comparison, full summary, targeted summary, example-based application, and internal logic explanation. To ensure a balanced distribution across these categories, the priority order of the generation prompts was randomized. We present the data distribution in Figure 2.

**Long-Form QA without References**   Recognizing the scarcity of training data for long-form QA without a provided reference—despite existing benchmarks (Wei et al., 2024b; Min et al.,

2023)—we developed a new dataset by repurposing the TriviaQA training set through the following meticulous pipeline:

1. **Question Selection:** We first identified questions where our baseline model (MiMo-7B-0530) exhibited partial but incomplete knowledge, selecting those it answered correctly in some, but not all, of eight sampling attempts.

2. **Reference Filtering:** We then filtered the accompanying reference documents (i.e., search results from the TriviaQA dataset) to a combined length of 500 to 60,000 characters. This ensured the context was sufficiently informative for validation without being computationally prohibitive for reward modeling.

3. **Question Generation:** Finally, we prompted an LLM to synthesize a new, open-ended question based on the filtered documents. Critically, during training, these source documents were withheld from the model being trained and were used exclusively by the validation model to verify the answer's faithfulness.

## 3.2 RL Algorithm

Building upon the model's strong foundational capabilities, we proceeded directly with reinforcement learning. We employ a variant of the GRPO algorithm, following the methodology of Yu et al. (2025); Xiaomi et al. (2025). Specifically, our implementation enhances the standard GRPO framework with several modifications, including the removal of the KL divergence penalty, the integration of Dynamic Sampling, and the application of a Clip-Higher mechanism.

## 3.3 Reward Modeling

We employ distinct reward functions for short-form and long-form QA to address their unique characteristics.

**Short-form QA** Following prior work by Song et al. (2025); Yang et al. (2025b); Xu et al. (2024); Yang et al. (2024), we use the following rule-based reward function:

$$f(y, y^*) = \begin{cases} -0.2, & \text{if the response format is incorrect (extraction failed)}, \\ 0.1, & \text{if } y \text{ constitutes a refusal to answer (e.g., "I don't know")}, \\ 1, & \text{if the response } y \text{ exactly matches the ground truth } y^*, \\ 0, & \text{otherwise.} \end{cases} \quad (1)$$

**Long-form QA** For long-form QA, our methodology is inspired by Jacovi et al. (2025); Wei et al. (2024b); Min et al. (2023). We utilize an LLM-as-a-judge approach where the model's response is decomposed into a set of atomic claims, each of which is then independently verified. Our composite reward function is defined as:

$$f(y) = f_{claim} - \alpha p_{format} - \beta p_{information\_density} \quad (2)$$

where the hyperparameters $\alpha$ and $\beta$, which weight the penalties, are both set to 0.2, respectively. The components are defined as follows:

**Factual Accuracy ($f_{claim}$):** A binary score for factual correctness. $f_{claim}$ is set to 1 if and only if all constituent claims are verified as supported by the reference material; otherwise, it is 0. Both claim extraction and verification are performed by an LLM.

**Format Penalty ($p_{format}$):** A term that penalizes formatting errors. The score is assigned by an LLM and is set to 1 if the output contains issues such as meaningless repetition or garbled text, and 0 otherwise.

**Information Density Penalty ($p_{information\_density}$):** A penalty score based on the relevance and density of information, assessed by an LLM against the reference. A higher score indicates a greater penalty for lower information density, assigned on a three-tier scale:

- A score of 1.0 for a response that sufficiently answers the question with no extra details.

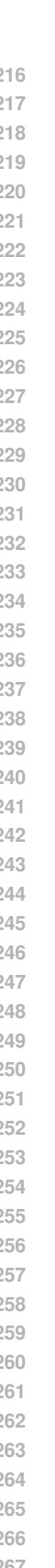
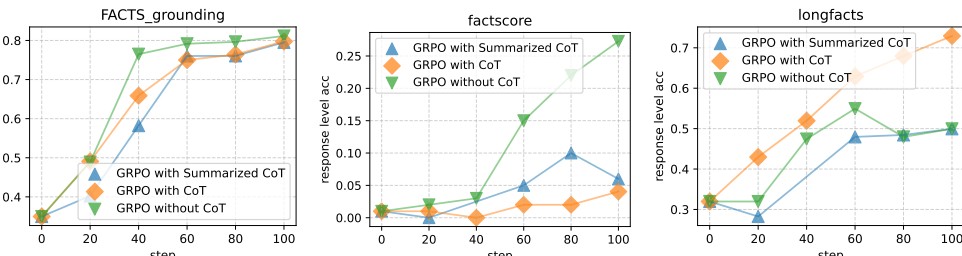

Figure 3: Performance Comparison of CoT Supervision Strategies Across Three Benchmarks.

- A score of 0.5 for a response that provides the answer with some additional, relevant information.
- A score of 0.0 for a response that offers a rich and comprehensive answer.

The LLM judge used for all reward scoring and verification tasks in our training process is GPT-OSS-120B (Agarwal et al., 2025).

## 4 EXPERIMENTS

### 4.1 SETUP

**Models**    We evaluate our proposed methodology on two open-source language models: MiMo-7B-RL-0530 (Xiaomi et al., 2025), and Qwen3-4B (Yang et al., 2025a).

**Benchmarks**    To rigorously evaluate the model's performance in mitigating hallucinations, we assess it on a comprehensive suite of benchmarks categorized by task format.

- **Unanswerable QA:** We use the Self-Aware dataset from Yin et al. (2023) and the Synthetic Unanswerable Math (SUM) test set from Song et al. (2025) to measure the model's ability to recognize and refuse unanswerable questions.
- **Short-Form QA:** We evaluate factual and reasoning accuracy on standard benchmarks, including AIME (MMA., 2024; 2025), TriviaQA (Joshi et al., 2017) and SimpleQA (Wei et al., 2024a).
- **Long-Form QA with Reference:** We use the Facts Grounding benchmark (Jacovi et al., 2025) to assess faithfulness to provided context. For this evaluation, we utilize the public test set, and all claims are verified using GPT-OSS-120B (Agarwal et al., 2025).
- **Long-Form QA without Reference:** To test for extrinsic (knowledge-based) hallucinations, we use 100 samples from FactScore (Min et al., 2023) and LongFact (Wei et al., 2024b) respectively. For these benchmarks, response verification is conducted using Gemini-2.5-Pro (Comanici et al., 2025) with search grounding.

**Metrics**    We employ the following metrics to provide a multi-faceted evaluation of model performance:

- **Response-Level Accuracy (Acc.):** For short-form QA, the criterion is a direct match with the reference answer. For long-form QA, scored as 1 only if all claims within a single response are factually correct; 0 otherwise. Notably, if the model refuses to answer in a long-form QA, an output with zero claims is also considered correct.
- **Claim-Level Accuracy (C. Acc.):** The proportion of individual claims across all test set responses that are factually correct. Responses are decomposed into atomic claims using an LLM prior to evaluation.
- **Average Claim Count (C. Num.):** The average number of atomic claims generated per response, measuring the model's information density.

- **Hallucination Rate (Hallu.):** This metric, based on the definition from Wei et al. (2024a), is calculated for benchmarks with ground-truth answers and represents the percentage of instances where the model provides a factually incorrect response.

## 4.2 SUPERVISING CHAIN-OF-THOUGHT REASONING

The question of whether to apply reward modeling to the intermediate steps of CoT reasoning is a subject of ongoing research. Prior work by Baker et al. (2025) found that direct supervision of CoT can yield modest performance gains but may also encourage models to develop undesirable "hacking" behaviors. To investigate this trade-off, we designed three distinct experimental conditions for our reward model:

**GRPO with CoT:** The entire reasoning chain is decomposed into atomic claims by an evaluator, and each claim is individually verified.

**GRPO without CoT:** Only the final answer is evaluated, while the CoT is ignored by the reward function.

**GRPO with Summarized CoT:** The model is prompted to first generate a CoT and then condense its reasoning into a concise summary, which is then submitted to the evaluator for verification.

Our findings indicate that full CoT supervision is computationally expensive and yields inconsistent results. As shown in Figure 3, while this approach improves performance on reference-based benchmarks like Facts Grounding, it degrades performance on open-domain tasks such as LongFact. This negative impact may arise because the evaluator misinterprets tentative or self-corrected steps within the reasoning chain as final, incorrect claims, thereby providing a misleading training signal.

Conversely, forgoing CoT supervision entirely leads to strong performance on LongFact but offers only marginal gains on Facts Grounding and FactScore. The summarized CoT approach, however, provides a more balanced outcome, achieving competitive performance across multiple benchmarks. Given that full supervision fails to deliver universal improvements and incurs substantial computational costs, we only adopted the summarized CoT supervision strategy for subsequent experiments. This method effectively balances the need to encourage sound reasoning while avoiding the pitfalls of over-penalizing the model's intermediate thought processes.

## 4.3 MAIN RESULTS

Our reinforcement learning methodology yields substantial and robust improvements across a wide range of tasks, as detailed in Table 1 and Table 2. The RL-tuned models demonstrate a significantly enhanced ability to refuse unanswerable questions, with accuracy on the Self-Aware and SUM benchmarks increasing to 79+ points for Qwen3-4B. This newly acquired cautiousness, trained on short-form data, successfully generalizes to more complex long-form scenarios. For instance, human evaluation revealed that our trained Qwen3-4B models learned to refuse approximately 30% of

| Method | Unanswerable | | short-QA | | | | AIME24 | AIME25 |
|---|---|---|---|---|---|---|---|---|
| | **Self-Aware** | **SUM** | **TriviaQA** | | **SimpleQA** | | | |
| | Acc.↑ | Acc.↑ | Acc.↑ | Hallu.↓ | Acc.↑ | Hallu.↓ | Acc.↑ | Acc.↑ |
| *MiMo-7B-RL-0530* | | | | | | | | |
| Base | 71.1 | 40.6 | 36.5 | 61.5 | 1.9 | 51.1 | 49.2 | 39.2 |
| Without CoT | 92.6 | 82.0 | 43.3 | 26.9 | 1.9 | 12.4 | 66.6 | 49.2 |
| with Sum. CoT | 85.9 | 84.4 | 45.9 | 27.9 | 2.2 | 14.2 | 65.0 | 52.5 |
| *Qwen3-4B* | | | | | | | | |
| Base | 53.9 | 5.5 | 33.7 | 65.9 | 2.7 | 70.6 | 62.5 | 50.0 |
| Without CoT | 93.4 | 78.1 | 41.2 | 18.8 | 0.7 | 5.0 | 55.0 | 43.3 |
| with Sum. CoT | 96.5 | 74.2 | 45.4 | 28.9 | 1.8 | 13.0 | 62.5 | 47.5 |

Table 1: Performance on short-form QA, unanswerable questions, and mathematical reasoning. "Hallu." denotes the hallucination rate. Evaluations for Self-Aware and SUM only use unanswerable subsets.

| Method | Facts grounding | | | FactScore | | | LongFact | | |
|---|---|---|---|---|---|---|---|---|---|
| | Acc. | C. Acc. | C. Num. | Acc. | C. Acc. | C. Num. | Acc. | C. Acc. | C. Num. |
| *MiMo-7B-RL-0530* | | | | | | | | | |
| Base | 35.0 | 85.2 | 14.2 | 1.0 | 19.1 | 19.7 | 32.0 | 84.7 | 23.3 |
| Without CoT | 82.3 | 95.4 | 6.2 | 21.2 | 32.7 | 9.6 | 43.0 | 90.5 | 15.9 |
| with Sum. CoT | 79.4 | 94.0 | 5.4 | 10.0 | 27.2 | 11.3 | 51.5 | 91.2 | 15.9 |
| *Qwen3-4B* | | | | | | | | | |
| Base | 47.1 | 83.6 | 8.5 | 5.0 | 28.7 | 16.6 | 38.0 | 89.9 | 22.6 |
| Without CoT | 82.4 | 92.6 | 4.5 | 44.0 | 52.2 | 5.2 | 74.7 | 96.2 | 11.2 |
| with Sum. CoT | 79.4 | 93.1 | 4.2 | 44.2 | 75.7 | 8.0 | 73.0 | 97.3 | 14.7 |

Table 2: Performance on long-form QA benchmarks. RL-tuned models show significant gains in response-level (Acc.) and claim-level (C. Acc.) accuracy, often accompanied by a decrease in the average number of claims (C. Num.).

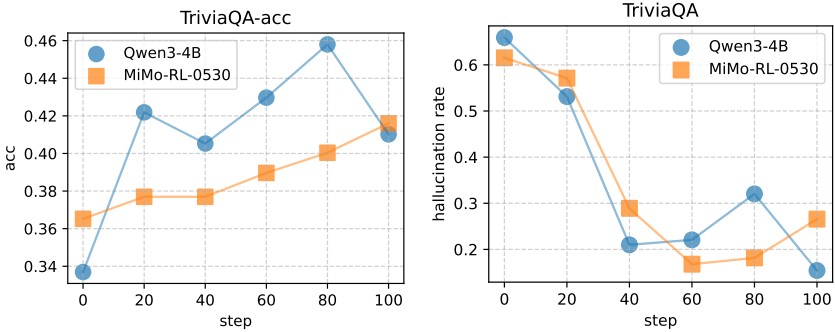

Figure 4: Training trajectory on TriviaQA. For MiMo-7B-RL-0530, the hallucination rate drops quickly and saturates early in training, after which accuracy begins to climb steadily.

questions (20% for MiMo) on the challenging FactScore benchmark rather than generating unsupported claims. This general reduction in confabulation is further evidenced by the precipitous drop in the Hallucination Rate on TriviaQA and SimpleQA. Crucially, these substantial gains are not model-specific; they are consistently observed on both the Mimo and Qwen3 models, underscoring the general applicability of our approach.

While the gains in reliability are clear, a more nuanced analysis reveals two important trade-offs. First, there is a distinct trade-off between factual accuracy and verbosity in long-form generation. While our methods dramatically increase both response-level and claim-level accuracy, this is consistently accompanied by a reduction in the average number of claims, indicating that the models learn to be more concise to remain factual. Second, we observe a task-specific performance trade-off in mathematical reasoning. While Mimo's AIME score improved significantly after tuning, Qwen3's performance notably declined. We hypothesize this regression stems from a data quality disparity, where our training data may be of lower quality than the proprietary data used in Qwen3's original alignment, inducing a degree of catastrophic forgetting.

## 5 ANALYSIS AND DISCUSSION

### 5.1 ASYMMETRIC LEARNING DYNAMICS: CAUTIOUSNESS VS. CORRECTNESS

A observation from our training process is the asymmetric learning dynamic between acquiring cautious behavior and improving factual correctness. We found that the model rapidly learns to adopt a refusal policy, such as responding with "I don't know" or identifying a question as unanswerable. In contrast, the acquisition of new knowledge or the refinement of complex reasoning patterns is a comparatively slower and more difficult process.

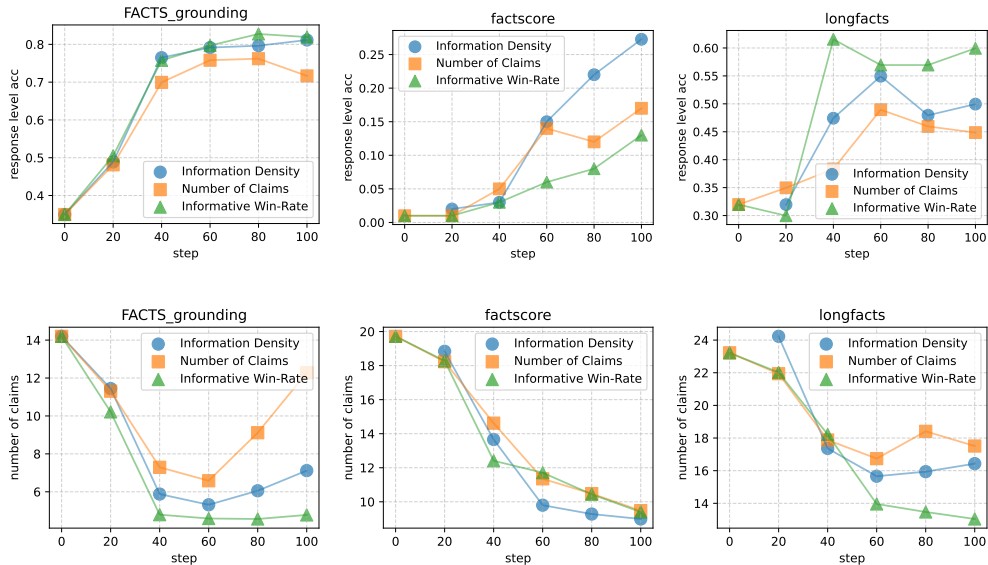

Figure 5: Comparison of Penalty Functions for Balancing Verbosity and Accuracy. The training dynamics illustrate that directly penalizing for a low number of claims can increase model verbosity late in training, but this explicitly compromises accuracy. The LLM and win-rate penalties achieve a more stable, albeit more concise, performance.

This disparity in learning rates is clearly illustrated by the training trajectory shown in Figure 4. During the initial training phase (before step 40), the model's hallucination rate drops precipitously, while its response accuracy increases only marginally. A significant rise in accuracy is observed only after the hallucination rate has stabilized at a lower bound, suggesting that the model first learns to stop providing incorrect information before it learns how to generate correct answers.

## 5.2 THE NECESSITY OF EXPLICIT REFUSAL INSTRUCTIONS

We investigated whether a model, trained with explicit refusal instructions, could generalize this cautious behavior to prompts where such instructions are omitted. To test this, we fine-tuned the model on a dataset where 50% of the training prompts included an explicit instruction to respond with "I don't know" for unanswerable questions, while the other 50% lacked this guidance.

During evaluation, the refusal instruction was removed from all test prompts. The model completely failed to generalize this capability, with its accuracy on the unanswerable questions dropping to zero. This result indicates that the model's ability to refuse an answer is tightly coupled to the presence of the explicit instruction and does not emerge as a generalized reasoning skill from this training setup.

Based on our findings, we recommend that to reliably enable cautious refusal capabilities in practical applications, developers should embed explicit instructions (e.g., "If the answer is unknown, state that you do not know") within the system prompt.

## 5.3 BALANCING FACTUAL ACCURACY AND INFORMATION DENSITY

A common failure mode in our experiments is reward hacking, where the model learns an evasive strategy: minimizing errors by providing overly brief or uninformative answers. To counteract this, we explored two more distinct approaches designed to encourage higher information density without reintroducing hallucinations.

**Informative Win-Rate:** The current model's output is compared against the output from the pre-RL, baseline model. An LLM judge selects the superior response, and a reward of +1 is given only if the current model's output is preferred with more information densities, ignoring the accuracy. This

method incentivizes the model to improve upon the baseline's overall quality, not just its factual accuracy.

**Number of Claims:** We calculate a score as the ratio of the number of claims in the current response to the number of claims in the baseline response. This creates a natural penalty for outputs that are less detailed than the original, with the score ranging from 0 to 1.

As illustrated in Figure 5, our experiments reveal a distinct trade-off between claim count and factual accuracy. The comparative win-rate method failed to meaningfully increase the number of claims. We hypothesize that this is because the verbose outputs of the baseline model contained a high frequency of hallucinations, causing the LLM judge to consistently prefer the RL-tuned model's shorter, more accurate responses.

Conversely, the claim count ratio method successfully increased the model's verbosity. However, this came at a significant cost to factual accuracy, with this approach yielding the lowest accuracy scores on the Facts Grounding, LongFact, and FactScore benchmarks. Neither approach perfectly resolved the tension between accuracy and information density. This suggests that the choice of penalty depends on the specific application's preference: prioritizing either maximal accuracy with concise outputs or accepting a higher risk of hallucination for more detailed responses.

# 6 LIMITATION AND FUTURE WORK

Our study, while comprehensive, has several limitations that open avenues for future research.

First, a key limitation is the dependency on a single LLM (GPT-OSS-120B) as the primary evaluator. The choice of the reward model can significantly influence training outcomes, a factor we did not systematically investigate. Our preliminary experiments using Gemini-Flash, for instance, yielded worse results compared to GPT-OSS-120B, which, despite its own tendency to hallucinate, proved to be a stricter and more effective judge for reference-grounded tasks. Additionally, the sufficiency of the judgment accuracy for smaller reward models warrants further exploration.

Furthermore, the diversity of our training data for long-form QA without reference tasks is constrained, as it is primarily derived from the TriviaQA dataset. This narrow data sourcing may restrict the model's generalization capabilities across different knowledge domains and question styles.

Finally, our current approach treats unanswerable queries in a binary fashion—the model either responds or refuses. A more sophisticated implementation would involve calibrating the model's confidence. An elegant extension would be to train the model to modulate its tone based on its certainty: adopting a firm tone for high-confidence answers, a tentative one for moderately confident responses, and refusing to answer when confidence is low. We believe this is achievable through a carefully designed reinforcement learning framework, contingent upon developing a robust reward function that can accurately quantify response confidence.

# 7 CONCLUSION

In this work, we present a comprehensive investigation into the use of RL to mitigate hallucinations in large language models across a variety of query types: unanswerable, short-form, and long-form. By synthesizing novel training datasets from sources like TriviaQA and FineWeb, we developed and evaluated targeted reward mechanisms for tasks with and without provided reference contexts.

Our key findings offer practical guidance for RL-based alignment. We demonstrate that directly rewarding the model's CoT process yields diminishing returns relative to its high computational cost. Furthermore, we identify and address a critical failure mode where models learn to evade hallucinations by reducing their information density, and we propose countermeasures to balance factual accuracy with response quality. Our proposed methodology demonstrates strong generalization, improving performance across multiple benchmarks and two different base models. Notably, we also show that training a model to cautiously refuse unanswerable questions is a straightforward process that does not negatively impact its performance on other tasks.

## ETHICS STATEMENT

The data we utilized are open for research, and evaluated LLMs are all publicly available by either parameters or API calls. All human evaluations mentioned in this paper are performed by the authors. Therefore, we do not anticipate any ethical concerns in our research.

## REPRODUCIBILITY STATEMENT

Our training framework is built upon verl (Sheng et al., 2024), vllm (Kwon et al., 2023), and megatron-core (Shoeybi et al., 2019). All experiments are conducted on a cluster of 32 Nvidia A100 GPUs. Our synthetic training data and the scripts for our reward function are available in the supplementary material. (Due to file size limitations, the training set we constructed from Fineweb will be provided after the review process is complete.) All other training data is sourced from public, open-source datasets. Detailed training hyperparameters can be found in the Appendix.

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

# APPENDIX

## A    USE OF LLM

The authors only used an LLM to polish the language of this paper.

## B    EXPERIMENT DETAIL

In our experiment, we employed a training batch size of 256. We use learning rate of 1e-6. The maximum sequence length was set to 32000 tokens to facilitate complex reasoning tasks. During the training phase, both temperature and top-p parameters were configured at 1.0 to promote output diversity. We applied on-policy GRPO for 140 steps for the results in Table 1 and 2.

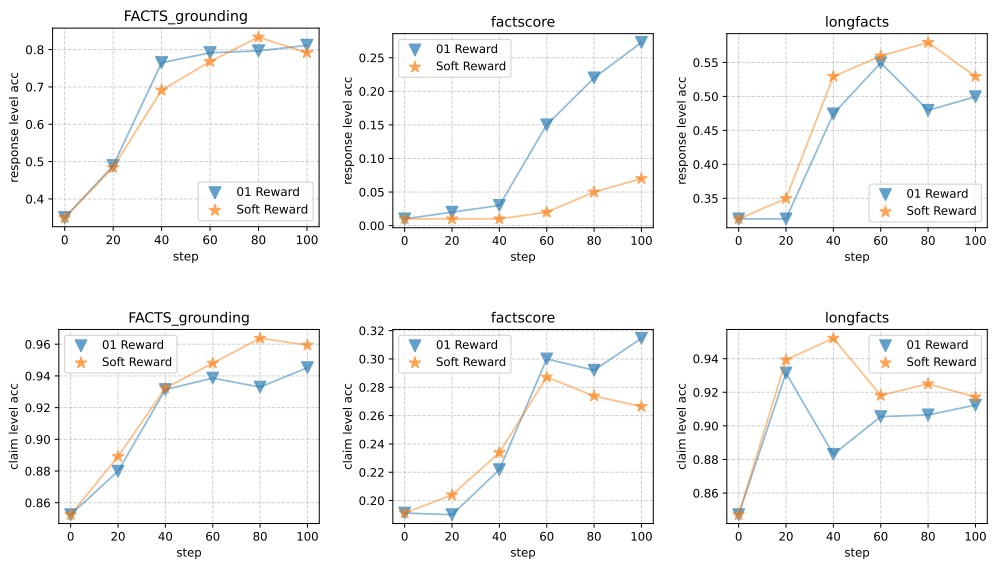

Figure 6: Training dynamic with different reward.

## C    SOFT REWARD VS. 0/1 REWARD

We also compared the performance of a soft reward signal against a binary (0/1) reward. The soft reward, defined by the formula $f_{claim} = \frac{N_{supported}}{N_{total}}$ (Equation 2), demonstrated a marginal advantage in claim-level accuracy (see Figure 6). This advantage was limited at the response level, particularly when measured by FactScore. For this reason, we did not elaborate on this difference.

## D  PROMPT FOR GENERATE LONG-FORM QA WITH REFERENCE

```
**1. Overall Task**
Analyze the source text provided below and generate a high-quality question based on the specified task types,
    rules, and priority.
The generated question must be fully answerable using only the information within the provided 'Source Text';
    no external knowledge should be required.

**2. Available Task Types**
* **Impact Analysis:** Asks about the subsequent impact or results of a key event, decision, or discovery
    mentioned in the text.
* **Internal Logic Explanation:** Asks about the underlying reasons and logic behind a rule, motivation, or
    design described in the text.
* **Example-based Application:** Asks for the creation of a specific case to demonstrate how an abstract
    concept or rule operates.
* **Specific Content Comparison:** Asks for a comparison of the similarities and differences between two or
    more related concepts, figures, or data points from the text.
* **Targeted Summary:** Asks for a precise, condensed summary of a specific sub-topic within the text.
* **Full Summary:** Asks for a general overview of the core ideas and conclusions of the entire text.

**3. Priority is:**
Impact Analysis > Internal Logic Explanation > Example-based Application > Specific Content Comparison >
    Targeted Summary > Full Summary

**4. Additional Generation Rules:**
* **Question Number:** The generated question can combine 1-3 different task types.
* **Length Limitation:** For summary tasks ('Targeted Summary', 'Full Summary'), the question can specify a
    word count limit, sentence limit or sentence limit (e.g., "in no more than X words" or "in at least X
    words").

**5. Source Text:**
{document}

**6. Return Requirement:**
Please return the result strictly in the following JSON format, without any additional explanations or text.
    When a question combines multiple task types, the '"Task Type"' field should reflect the one with the
    highest priority.

```json
{{
  "Source Text": "This should contain the complete source text you provided in point 5 above",
  "Task Type": "This should be the name of the highest-priority task type selected from point 2, for example:
      'Impact Analysis'",
  "Generated Question": "This should be the final, specific question string that was generated"
}}
```
```

# E  PROMPT FOR GENERATE LONG-FORM QA WITHOUT REFERENCE

```
Based on the document(s) provided below, rewrite the "Original Question" into a single, open-ended question.

Guidelines for the rewritten question:

For a person or thing: It could ask for an introduction, significant experiences, major impacts, key honors,
    etc.

For an event: It could ask about its causes, timeline, key people/things involved, and its resulting
    consequences.

For a concept or theory: It could ask about its origins and development, the key figures who advanced it, its
    influence, and practical application examples.

Crucial Requirement: You must ensure that the rewritten question can be answered and fully verified using only
        the information given in the document(s). The only exception is for answers that can be derived by pure
        reasoning based on the provided text.
It should be noted that the respondents cannot see these documents, so please do not mention phrases similar
        to "answer based on the references" in the questions.

Example
Document(s):
(Assume the documents contain information about Rafael Nadal's victory at the 2008 Wimbledon final, his
    overall career, and his well-known dominance on clay courts, which led to his nickname.)

Original Question:
Who won the 2008 Men's Singles Final at Wimbledon?

Rewritten open_ended question:

```open_question
Introduce Rafael Nadal and explain why he is known as "The King of Clay".
```

Format your final response as a single code block as shown in the example above.

-----

Document(s):
(The text may contain multiple documents separated by #####)
{document}

Original Question:
{question}

Original Answers:
{answer}
```

# F  PROMPT FOR CLAIM-LEVEL REWARD

```
# Role & Goal

You are a helpful and harmless AI assistant. You will be provided with a textual context and a model-generated
        response.
Your task is to analyze the response sentence by sentence and classify each sentence according to its
        relationship with the provided context.
Generate a single, comprehensive JSON object that summarizes the response's quality across multiple dimensions
        inside json block.

**Input Format:**

The input will consist of two parts, clearly separated:

* **Context:**    The textual context used to generate the response.
* **User Query:** The question raised by the user regarding the context.
* **Response:**   The model-generated response to be analyzed.

# Instructions

Your final output **must be a single JSON object inside json block**. Follow the steps and definitions below
        to construct this object.

**Step 1: Sentence-by-Sentence Analysis**
First, break down the 'Response' into individual sentences. For each sentence, perform an analysis and store
        the results in a list named 'sentences_check'. Each object in this list must contain:

  - 'sentence': (string) The original text of the sentence.
  - 'label': (string) One of the following four labels:
      - **'supported'**: The sentence is directly entailed by the given 'Context'.
      - **'unsupported'**: The sentence is not entailed by the given 'Context'.
      - **'contradictory'**: The sentence is falsified by the given 'Context'.
      - **'no_rad'**: The sentence does not require factual attribution (e.g., opinions, greetings, questions,
            disclaimers).
  - 'rationale': (string) A brief explanation for the assigned label.
  - 'excerpt': (string) A direct quote from the 'Context'. This is **required** for 'supported' and '
        contradictory' labels and should be 'null' otherwise. The excerpt must fully support or contradict the
        sentence.

**Be extremely strict:** Unless you can find a clear, indisputable excerpt, default to 'unsupported'. Do not
        use world knowledge.

**Step 2: Generate Top-Level Metrics**
After completing the sentence analysis, create the following top-level keys in the JSON object:

  - 'overall_reasoning': (string) A global summary explaining your final evaluation and the reasoning behind
        the key metric scores.

  - 'has_formatting_errors': (boolean) Set to 'true' if the response has issues like meaningless repetition,
        truncation, garbled text, multiple '<think>' tags, or any format errors. Otherwise, set to 'false'.

  - 'all_sentences_grounded': (boolean) Set to 'true' **if and only if** all sentences in the 'sentences_check
        ' list are labeled either 'supported' or 'no_rad'. If any sentence is 'unsupported' or 'contradictory
        ', set this to 'false'.

  - 'request_completed': (boolean) Set to 'true' if the response fully and correctly addresses all parts of
        the 'User Query', including any constraints like word count, sentence count, or tone. Otherwise, set
        to 'false'.

  - 'completeness_score': (integer, 0-2) A score for the quality of the response and its reasoning, based on
        the following scale:

      - **0**: Answered the question but provided no explanation.
      - **1**: Provided some explanation, but it was not coherent, detailed enough, or was overly verbose.
      - **2**: Provided a reasonable response with a complete, clear, and concise explanation.

# Example

**Input:**

```
Context: Apples are red fruits. Bananas are yellow fruits.

User Query: Tell me some thing about apples and bananas.

Response: Apples are red. Bananas are green. Bananas are cheaper than apples. Enjoy your fruit!
```

**Output:**

```json
{{
    "sentences_check": [
        {{
            "sentence": "Apples are red.",
            "label": "supported",
            "rationale": "The context explicitly states that apples are red.",
            "excerpt": "Apples are red fruits."
        }},
```
```

```
        {{
            "sentence": "Bananas are green.",
            "label": "contradictory",
            "rationale": "The context states that bananas are yellow, not green.",
            "excerpt": "Bananas are yellow fruits."
        }},
        {{
            "sentence": "Bananas are cheaper than apples.",
            "label": "unsupported",
            "rationale": "The context does not mention the price of bananas or apples.",
            "excerpt": null
        }},
        {{
            "sentence": "Enjoy your fruit!",
            "label": "no_rad",
            "rationale": "This is a general expression and does not require factual attribution.",
            "excerpt": null
        }}
    ],
    "overall_reasoning": "The response correctly identified one fact but contradicted another and introduced
        an unsupported claim. Therefore, it is not fully grounded in the context.",
    "has_formatting_errors": false,
    "all_sentences_grounded": false,
    "request_completed": true,
    "completeness_score": 0
}}
```

**Now, please analyze the following context and response:**

**Context:**
{context_document}

**User Query:**
{user_request}

**Response:**
{response}
```