# OpenReview forum: "Enhancing Reliability across Short and Long-Form Question Answering via Reinforcement Learning"
_ICLR.cc/2026/Conference — ICLR 2026 Conference Withdrawn Submission_

### Official Review · Reviewer_2QkL · 2025-10-30

**Soundness:** 2
**Presentation:** 2
**Contribution:** 2
**Rating:** 4
**Confidence:** 4

**Summary:**

This paper presents a targeted RL framework to mitigate both intrinsic and extrinsic hallucinations in LLMs for short and long-form question-answering tasks. The authors make several contributions, including the development of novel training datasets and a thorough empirical analysis of various training strategies. Despite the novelty of the application scenario, the work is not solid enough, especially the presence of several serious typology issues which leads me to believe this work requires significant further effort.

**Strengths:**

1. The paper primarily trains the LLM's refusal capability in short-form QA to enhance reliability and successfully generalizes this to long-form QA scenarios, which is a novel and valuable exploration of the setting.
2. The paper presents some interesting insights through its experiments, such as the asymmetry in learning dynamics (the model learns to refuse much faster than it learns to be correct).

**Weaknesses:**

1. Lack of baselines. Comparing only against the base model makes it difficult to verify that the proposed method is indeed effective compared to existing methods. There are many existing alignment works in the direction of LLM reliability, such as [1][2].
2. Insufficient analysis of the post-trained model. The analysis in this work focuses on experimental settings and additional insights, while the advantages of the method are only shown in the main table and are not concretely demonstrated. For example, there is no specific analysis on whether the model selectively refuses to answer difficult questions, the distribution of refusal behavior across datasets of varying difficulty, or the generalization of refusal behavior trained on short-form QA to long-form QA.
3. Reliance on an LLM Judge with limited capabilities. You mentioned the potential negative impact of the LLM judge's performance on the results in several places. For instance, the paper mentions that "the evaluator may provide a misleading training signal." It also notes that "the LLM judge consistently prefer the RL-tuned model’s responses" over the baseline model, but in the Informative Win-Rate setup, the focus should have been on information density while ignoring accuracy. These errors raise concerns about the validity of the training results and the analytical conclusions.
4. Under-investigated Catastrophic Forgetting. The RL-tuned Qwen3-4B model showed a notable performance decline on the AIME mathematical reasoning benchmark. The analysis of this phenomenon is insufficient. Exploring why this occurred and potential mitigation strategies would have strengthened the paper.
5. There are several serious and non-negligible typographical errors.
    1. The observations and conclusions in Figure 3 are inconsistent with the text in section 4.2. The text states that full CoT improves performance on Facts Grounding but degrades performance on LongFact. However, the chart shows that on Facts Grounding, the accuracy of full CoT (orange line) is lower than that of without CoT (green line), while on LongFact, full CoT is significantly better than the other methods. Figure 3 is in complete conflict with the observations and conclusions.
    2. The content on line 283 is incomplete. The sentence "To improve the model’s ability to recognize unsolvable queries, 25" is cut off.
    3. The paper alternates between "Mimo" and "MiMo". It is recommended to maintain consistent capitalization.

References
[1] https://arxiv.org/abs/2311.09677

[2] https://arxiv.org/abs/2403.18349

**Questions:**

1. In the CoT experiments in section 4.2, you mentioned that the poor performance of Full CoT might be because "the evaluator misinterprets tentative or self-corrected steps within the reasoning chain as final, incorrect claims." However, this could be avoided by simply using an LLM to extract the final answer before evaluation. I suggest you could try this approach.
2. In section 5.2, you demonstrate that the model fails to generalize its refusal capability without explicit instructions in the prompt. Have you tried to guide the model to generate refusal responses without including any explicit refusal instructions in the training data? This approach was used in many papers like [1] and was shown to work. Maybe you could give it a try.
3. In your long-form QA reward function, the hyperparameters alpha (format penalty) and beta (information density penalty) were both set to 0.2. How were these values determined? Was any sensitivity analysis performed to understand their impact on the training trade-offs?

References

[1] https://arxiv.org/abs/2311.09677

---

> ### Author Response · Authors · 2025-11-17
>
> We sincerely appreciate the reviewers' comments and suggestions, and would like to provide clarifications on the following points:
>
> ## Baseline
>
> We do not intend to emphasize our contributions to the "refusal" aspect (as the related training set is from open source); instead, we regard it merely as part of supporting our core goal of achieving "reliability".
>
> Our primary contributions focus on reinforcement learning training for long contexts and the construction of the training dataset. We are currently using WildChat as a baseline to further validate the effectiveness of our method.
>
> ## Insufficient Analysis of the Post-Trained Model
>
> We sincerely thank the reviewer for pointing this out.
> - Regarding refusal on difficult questions: Similar conclusions are presented in Section 4.3 and Figure 4, where the model refuses to answer questions that it cannot solve (i.e., more difficult questions). As shown in Table 1, the reduction in hallucination (from 0.66 to 0.19) is more significant than the increase in accuracy (from 0.34 to 0.41), implying that the model refused 40% of difficult questions. We will add more descriptions in subsequent revisions to highlight this conclusion.
> - Regarding generalization from short texts to long texts: As mentioned in Section 4.3, the majority of the improvement in FactScore of our model stems from the generalization of refusal. Since small models lack the necessary knowledge to answer the questions in FactScore, they can only gain scores by refusing to answer—and all these results have been verified by human annotators (relevant descriptions are available in the Ethics Statement). We will further emphasize this conclusion in the revised version.

---

> ### Author Response · Authors · 2025-11-17
>
> ## Reliance on an LLM Judge with Limited Capabilities
> It is true that the judgment ability of LLM judges is limited. However, in practical training, we could not find a better alternative for evaluating long-text answers. Therefore, we attempted to optimize the questioning method to achieve comprehensive auto-evaluation—and in fact, we did obtain improved results through this approach.
> ## Under-Investigated Catastrophic Forgetting
> We have not conducted a more thorough analysis of this phenomenon because its occurrence is inconsistent across different models. In fact, as shown in Table 1, mathematical ability is not lost in most cases; only Qwen3-4B (without CoT) exhibits a significant performance drop. Setting aside the randomness of evaluation, we believe that completely abandoning supervised thought paths (i.e., CoT) may lead to certain performance losses. Thus, we suggest incorporating simplified CoT evaluations in practice (e.g., "with Summarized CoT"). We will clarify this point in future version.
> ## Typo Errors
> We sincerely apologize for this proofreading error caused by manual writing.
> 1. We will revise this description in subsequent versions, and the results presented in the figures and section 4.2 remain accurate.
> 2. This error arises from the incorrect use of the percentage symbol (%). We will fix this issue in the revisions.
> 3. We will adopt the standard notation of "MiMo".

---

> ### Author Response · Authors · 2025-11-17
> **Questions**
>
> ## Question 1
> Our "without CoT" setup is exactly implemented as described. After observing that this setup does not cause performance loss, we adopted a similar configuration in most of our experiments.
> ## Question 2
> We sincerely appreciate your suggestion. We are currently considering adding experiments where instructions are gradually removed to enhance the generalization ability of the model. We will update the results in a timely manner and revise the final manuscript accordingly.
> ## Question 3
> In some early experiments, we found that specific values have little impact on the results; thus, we did not conduct further exploration of this aspect afterward. If time permits, we will add the corresponding experiments to address this point.

---

### Official Review · Reviewer_1Q1w · 2025-11-01

**Soundness:** 3
**Presentation:** 3
**Contribution:** 2
**Rating:** 4
**Confidence:** 3

**Summary:**

This paper proposes a reinforcement learning (RL) framework aimed at mitigating both intrinsic and extrinsic hallucinations in Large Language Models (LLMs) across short and long-form Question Answering (QA) tasks. The core contributions are threefold:

1. A Novel RL Framework: An application of a GRPO-variant RL algorithm to jointly improve factual accuracy and encourage refusal on unanswerable questions.

2. New Training Datasets: The construction of two long-form QA datasets: one for intrinsic hallucination (using FineWeb-derived, reference-grounded Q&A) and one for extrinsic hallucination (using open-ended questions converted from TriviaQA).

3. Empirical Analysis: A series of experiments analyzing factors like Chain-of-Thought (CoT) supervision and the trade-off between factual accuracy and information density.

The authors demonstrate performance improvements on a suite of standard benchmarks (e.g., TriviaQA, Facts Grounding, LongFact) and provide insights into the learning dynamics of cautiousness versus correctness.

**Strengths:**

Comprehensive Scope: The paper tackles a broad and critical problem—mitigating both intrinsic and extrinsic hallucinations—across a wide range of QA formats. This holistic approach is more ambitious than prior work focused on a single facet.

High-Quality Empirical Evaluation: The experimental section is thorough, using multiple models (MiMo, Qwen) and a diverse set of well-chosen benchmarks to validate the claims from different angles.

Valuable Analysis: The investigation into CoT supervision (Section 4.2) yields a clear and practical finding: that full CoT supervision is not cost-effective for this task. The analysis of asymmetric learning dynamics (Section 5.1) and the accuracy-density trade-off (Section 5.3) provides nuanced insights beyond mere performance numbers.

Clarity of Core Method: The overall framework and the design of the reward functions for short and long-form QA are explained with sufficient clarity for the reader to understand the high-level approach.

**Weaknesses:**

Reproducibility Crisis: The heavy, non-ablatable dependence on multiple proprietary, massive LLMs (GPT-OSS-120B, Gemini-2.5-Pro) for both training and evaluation is a major weakness. It places the work outside the reach of most academic labs for reproduction or direct building upon, which is a significant concern for a scientific conference.

Lack of Ablation Studies: The paper presents a complex pipeline but does not isolate the effects of its key components. How much do the novel datasets contribute versus the reward design? What is the individual impact of removing the KL penalty? Without ablations, the source of the improvements remains ambiguous.

Insufficient Algorithmic Detail: As noted, Section 3.2 is unacceptably vague for a technical conference. The "variant of GRPO" must be described in precise detail, or the code must be released for the review to properly assess the algorithmic contribution.

Superficial Treatment of a Key Finding: The failure of instruction generalization (Section 5.2) is a critical and disappointing result, but it is not explored in depth. Why does this happen? Is it a limitation of the model size, the training data distribution, or the RL objective? A deeper discussion is needed.

**Questions:**

1. Reproducibility & Cost: Given the reliance on GPT-OSS-120B for reward modeling and evaluation, what was the total computational and financial cost of this project? Will the authors commit to releasing their synthesized training datasets and the exact prompts used for the LLM judge to ensure a degree of reproducibility for the community?

2. Ablation Study: Can the authors provide an ablation study to quantify the contribution of (a) the new long-form QA datasets, and (b) the individual terms in the long-form reward function (e.g., the information density penalty)?

3. Algorithmic Detail: Could the authors provide a detailed, formal description of their GRPO variant, specifically clarifying the "Dynamic Sampling" and "Clip-Higher" mechanisms, and justifying the removal of the KL divergence penalty?

4. Generalization of Refusal: The finding in Section 5.2 that refusal does not generalize without explicit instructions is significant. What are the hypothesized reasons for this failure? Did the authors experiment with alternative training strategies (e.g., gradually phasing out the instruction) to encourage generalization?

5. Statistical Significance: The results in Tables 1 and 2 represent a single run. Can the authors report results with standard deviations over multiple random seeds, or perform statistical significance tests to bolster the claims of improvement?

---

> ### Author Response · Authors · 2025-11-17
> **Reproducibility & Cost**
>
> In fact, only GPT-OSS-120B was extensively utilized during the training phase. In our study, we deployed GPT-OSS-120B using 2 local H100 GPUs, which helped control API costs.
>
> The use of Gemini-Pro was limited to evaluation, with each evaluation costing approximately $30.
>
> Taking a single experiment (for Qwen3-4B) as an example: we used 32 A100 GPUs for training/inference and 2 H100 GPUs for deploying GPT-OSS, and the entire experiment could be completed in approximately two days. Calculated at a rate of 2.25 dollars (H100) and 1.2 dollars (A100) per hour, the cost of each experiment is around 2,000 dollars.
>
> We acknowledge that this constitutes a considerable expense for the academic community, especially for other studies on hallucination/factual accuracy. Nevertheless, compared with other reinforcement learning (RL) approaches, our training costs remain relatively low.
>
> By the way, all the prompts have been provided in the Appendix.

---

> ### Author Response · Authors · 2025-11-17
> **Ablation Study and Algorithmic Detail**
>
> We sincerely appreciate the reviewer's suggestions and are actively conducting these additional experiments.
>
> We are preparing to conduct a vanilla GRPO experiment to verify its independence from specific methods.
>
> Additionally, we plan to perform experiments on WildChat to validate the effectiveness of our data.
>
> In fact, our model training method has only adhered to certain operations deemed effective for overall RL [1-2].
>
> We apologize for the omission of this description. These are commonly used techniques in other studies [1-2], and we provide a more detailed explanation here. In the final submitted version, if constrained by paper length, we will include these contents in the appendix.
>
> - Removal of KL Loss: Simply removing the KL loss effectively unleashes the full potential of the policy model without compromising training stability.
> - Dynamic Sampling: During the RL rollout phase, we over-sample prompts and filter out those with a pass rate of 1 or 0. This leaves all prompts in the batch with effective gradients while maintaining a consistent batch size, and the strategy automatically calibrates the difficulty of problems throughout policy training.
> - Clip-Higher: We increase the upper clip bounds in GRPO. This measure can mitigate the entropy convergence problem and facilitate the policy to explore new solutions.
>
> We have
>
> $$
> \mathcal{J}_{GRPO}(\theta) =
> \mathbb{E}_{q \sim D, \{o_i\}_{i=1}^G \sim \pi_\theta(\cdot|q)}
> \left[
> \frac{1}{\sum_{i=1}^G |o_i|}
> \sum_{i=1}^G \sum_{j=1}^{|o_i|}
> \min \Bigg(
> \frac{\pi_\theta(o_i|q)}{\pi_{\theta_\mathrm{old}}(o_i|q)} A_{i,j},
> \mathrm{clip}\Big(\frac{\pi_\theta(o_i|q)}{\pi_{\theta_\mathrm{old}}(o_i|q)},
> 1-\varepsilon_\mathrm{low}, 1+\varepsilon_\mathrm{high}\Big) A_{i,j}
> \Bigg)
> \right]
> $$
>
> where $\varepsilon_\mathrm{low}0.2$ and $\varepsilon_\mathrm{high}=0.27$
>
> [1] Yu, Qiying, et al. "Dapo: An open-source LLM reinforcement learning system at scale." arXiv preprint arXiv:2503.14476 (2025).
>
> [2] Xiaomi, L. L. M., et al. "MiMo: Unlocking the Reasoning Potential of Language Model—From Pretraining to Posttraining." arXiv preprint arXiv:2505.07608 (2025).

---

> ### Author Response · Authors · 2025-11-17
> **Generalization of Refusal and Statistical Significance**
>
> We believe the primary cause of failure is the lack of models’ response patterns for unanswerable questions during pre-training. For this reason, we consider the construction of SFT data to be relatively challenging and unnatural.
>
> If time permits, we will add experiments that gradually promote generalization.
>
> Due to the high cost of RL experiments, we have not yet conducted experiments with additional random seeds.
> If time allows, we will provide two more sets of results to enhance statistical significance.

---

> ### Comment · Reviewer_1Q1w · 2025-11-20
>
> I acknowledge and appreciate the authors' detailed response. However, in my view, the paper still has some inherent limitations in terms of methodological novelty or theoretical depth, which have not been fundamentally overcome by the rebuttal. Therefore, after careful consideration, I have decided to maintain my current score.

---

### Official Review · Reviewer_3JMt · 2025-11-02

**Soundness:** 2
**Presentation:** 2
**Contribution:** 2
**Rating:** 4
**Confidence:** 4

**Summary:**

This work considers three types of QA tasks: short-form QA, and long-form QA with and without references. To ensure good performance across all three tasks, the authors synthesized training data and conducted reinforcement learning training, ultimately providing several experimental insights.

**Strengths:**

* The paper is clearly written and easy to follow. Although the work leans toward an engineering contribution rather than deep theoretical novelty, it provides ample implementation details and experimental transparency, making it highly reproducible and accessible to practitioners.
* The authors carefully construct a new training corpus integrating both short- and long-form QA data (from FineWeb and TriviaQA), explicitly designed to address intrinsic and extrinsic hallucinations.

**Weaknesses:**

* Although the paper’s data construction process is well-executed and methodologically solid, all datasets are either synthetic or repurposed from existing sources (TriviaQA and FineWeb). As a result, the work does not introduce any genuinely new data or annotations, limiting its originality as a dataset contribution.
* The reinforcement learning experiments are relatively superficial. Key ablations, such as varying the reward weighting coefficients, analyzing the effect of removing or modifying specific reward components, or exploring different KL/divergence regularization schemes, are missing. These are essential to understand the robustness of the proposed reward formulation.
* The experiments primarily compare RL-trained models against their untuned base versions on the same dataset. This setup is insufficient to isolate the contribution of the newly constructed dataset or the RL pipeline itself. Comparisons against (1) SFT-only models trained on the same data, and (2) RL models trained on alternative open-source datasets, are necessary to demonstrate that the proposed dataset and framework provide unique benefits.
*
* typo: It seems that something is missing on line 151.

**Questions:**

* Why did you choose MiMo-7B and Qwen3-4B for their experiments?
In other words, why didn’t you conduct experiments using Qwen3-8B instead?

---

> ### Author Response · Authors · 2025-11-17
>
> We would like to express our sincere gratitude for your review comments and valuable suggestions. Herein, we provide responses to the raised questions as follows:
>
> 1. Why were MiMo-7B and Qwen3-4B selected instead of Qwen3-8B?
>
> As indicated in several reports [1-2], MiMo-7B and Qwen3-4B have been shown to exhibit significant hallucination issues. Furthermore, the official team of Qwen3 also regards the 4B version as their primary small-scale model for demonstration [3]. Regarding Qwen3-8B, its exclusion was due to constraints on experimental resources available at the time of our study.
>
> 2. The experimental data does not include any new data or annotations.
>
> We respectfully disagree with this statement. First, for the high-quality long texts from FineWeb, we generated targeted questions—with the guarantee that these questions can be answered by referring to the long texts—and this essentially constitutes a form of annotation.
>
> 3. Concerning the lack of experiments.
>
> We are actively conducting additional experiments and will definitely update the results before the end of the rebuttal period.
>
> [1] Yao, Zijun, et al. "Are Reasoning Models More Prone to Hallucination?." arXiv preprint arXiv:2505.23646 (2025).
> [2] Song, Linxin, Taiwei Shi, and Jieyu Zhao. "The hallucination tax of reinforcement finetuning." arXiv preprint arXiv:2505.13988 (2025).
> [3] https://qwen.ai/blog?id=qwen3

---

> ### Comment · Reviewer_3JMt · 2025-11-21
>
> We thank the authors for their detailed response and the additional clarifications. We appreciate the effort to address the concerns raised in the initial review. However, several important issues remain insufficiently resolved.
>
> ---
>
> ### On the choice of MiMo-7B and Qwen3-4B instead of Qwen3-8B
>
> The rebuttal argues that MiMo-7B and Qwen3-4B have been reported to exhibit significant hallucination issues, and that Qwen3-4B is the primary small-scale model used for demonstration by the Qwen team. While these points are reasonable, they do not address the core concern:
>
> The fact that MiMo-7B and Qwen3-4B hallucinate heavily does not imply that Qwen3-8B does not, nor does it justify omitting Qwen3-8B in an evaluation that aims to make general claims about hallucination mitigation.
>
> The current experimental setup evaluates two models of different sizes and from different sources. This makes it difficult to convincingly argue for the generalizability of the proposed method:
>
> * A comparison between Qwen3-4B and Qwen3-8B would allow the authors to draw conclusions about the behavior of the method across different model scales within the same family.
>
> * A comparison between Qwen3-8B and MiMo-7B would help demonstrate robustness across model families / sources.
>
> In other words, including Qwen3-8B is not just a matter of adding “one more model”; it is essential for supporting the claimed generality of the approach along two important axes: model size and model origin.
>
> Regarding the stated resource constraints: the rebuttal mentions that MiMo-7B could be run but Qwen3-8B could not, due to limited resources. Given that these models are of comparable size, this explanation is not very convincing. If the experimental setup is capable of handling MiMo-7B, it is reasonable to expect that running Qwen3-8B should be similarly feasible, or that a more detailed justification of the resource limitation be provided (e.g., differences in context length, inference setup, infrastructure constraints, etc.). As it stands, the omission of Qwen3-8B remains a notable weakness in the experimental design.
>
> ---
>
> ### On the data construction
> I acknowledge the data contribution and agree that generating question–answer pairs from high-quality long texts constitutes a form of annotation. However, in its current form, this contribution does not seem sufficient to support one of the core claims of the paper.
>
> In the introduction, the work places substantial emphasis on the training data as a key contribution. Yet, according to the rebuttal and the paper, the data is essentially constructed by combining existing data sources and annotations generated purely by existing large language models, without additional human annotation, careful manual curation, or a clearly distinctive data-generation pipeline.
>
> From a contribution perspective, this makes it difficult to view the dataset as a strong standalone novelty. Many recent works similarly construct synthetic QA or instruction data from existing corpora using LLMs, and the current description does not convincingly differentiate your data pipeline from this growing body of work

---

### Note · Authors · 2025-12-05

**Comment:**

We sincerely thank the reviewers for their insightful feedback and in-depth comments. Due to constraints on the authors' time and resources, and after careful consideration, we have decided to withdraw this manuscript. We are confident that this work will reappear in a much stronger form in the near future.

However, we would like to share some updates with the reviewers regarding the points raised:

1. Regarding the discussion on generalization without a system prompt: Through manual verification, we confirmed that the model effectively generalizes on the test set. The previous discrepancy arose because we directly reused the "exact match" evaluation script intended for the version with a system prompt. We have since switched to an "LLM-as-a-Judge" evaluation method and obtained new results: the probability of refusal in the condition without a system prompt is comparable to (or even slightly higher than) the condition with a system prompt. This is because the LLM-as-a-Judge method accepts various forms of refusal rather than being restricted to a specific format.

2. Regarding the comparative experiments: We successfully conducted the comparative experiments using WildChat. The evaluation results across three long-form QA benchmarks show that its performance is significantly inferior to our proposed method.

3. Regarding the remaining experiments: Due to the need to prioritize other recent projects, we were unable to allocate sufficient resources to conduct the remaining two promised experiments (the 8B model and the original GRPO). We sincerely apologize for this. We intend to include these results prior to our next submission.

Once again, we thank the reviewers for their valuable advice on our manuscript.

**Withdrawal Confirmation:**

I have read and agree with the venue's withdrawal policy on behalf of myself and my co-authors.